# Three-dimensional sculpting of laser beams

Tobias Kree[1], Michael Köhl[1]

**1** Physikalisches Institut, University of Bonn, Wegelerstraße 8, 53115 Bonn, Germany

July 3, 2020

## Abstract

**We demonstrate three-dimensional sculpting of laser beams using two-dimensional holograms. Without relying on initial guesses of the analytic properties or the Fourier transform of the desired light field, we show that an improved numerical phase retrieval algorithm can produce continuous three-dimensional intensity distributions of arbitrary shapes. We benchmark our algorithm against optical bottle beams and double-helix beams and then show the extension to complex optical structures.**

## 1 Introduction

Holographic beam shaping has developed into a powerful technique wherever laser light needs to be tailored to the special requirements of its respective application. The ability to engineer the spatial intensity profile of a light field has empowered novel and sophisticated methods of microscopy, optical trapping and optical manipulation. For example, absorptive microparticles have been confined in single-beam optical bottles [1] or colloidal spheres have been steered along curved trajectories with Airy beams [2]. Equally, beam shaping has served to achieve single-beam, three-dimensional imaging utilizing engineered point spread functions in super-resolution microscopy [3, 4].

However, the creation of advanced light fields with arbitrary three-dimensional intensity distribution remains a challenging problem. Commonly they are created from analytic solutions or closed-form expressions for the electric field (rather than the intensity), thereby restricting the set of realizable beams. For instance, the abruptly autofocussing beams derived from the Airy solution [5] can form three-dimensional structures [6] or even single-beam optical bottles [7]. These approaches have in common that either the exact desired optical field or its Fourier transform have to be known, which is much more restrictive than specifying the intensity distribution. Often this requires simplifying assumptions such as cylindrical symmetry [8,9] or an analytic mode basis [3]. Therefore, the properties of beams created with the aforementioned approaches are intrinsically limited.

Numerical approaches using iterative projection algorithms have already established arbitrary two-dimensional beam shaping with remarkable capabilities [10]. Extending the beam shaping to a finite volume requires three-dimensional beam sampling and constraint application. Sampling the volume on a three-dimensional grid allows to retrieve complex structures [11, 12] but extensive volumetric sampling comes along with a high computational load. Sampling only at multiple axially shifted target planes helps to reduce the problems complexity still allowing for highly versatile beam shaping [13]. Decreasing the axial sampling rate can result in an uncontrolled intra-plane beam propagation [13]. Retaining sufficient control over the intra-plane field propagation marks an important step towards efficient still highly versatile continuous beam shaping.

In this paper, we demonstrate spatially continuous three-dimensional intensity sculpting using an improved numerical phase retrieval method. The appeal of this approach is based on its overall simplicity while allowing for high flexibility. We show that our approach cannot only reproduce complex beams but it is even capable of modifying their beam profile during propagation in a predictable manner. We demonstrate our approach using the examples of a single-beam optical bottle [9] and a rotating double-helix point spread function [3] without providing any analytical input. We then show that the methodology can be extended beyond cylindrical symmetry and beyond simple scaling transformations.

## 2    Experimental setup and volumetric phase retrieval

The experimental setup (see Figure 1) is composed of a spatial light modulator at location $z = 0$, which is illuminated by a collimated Gaussian laser beam of waist $w_0$=6.3mm and a wavelength of $\lambda$=735nm. The phase-only spatial light modulator [14] imprints a phase pattern $\phi_{\mathrm{SLM}}$ onto the Gaussian beam. The beam after phase modulation is imaged by a thin lens ($f$=250mm) in a $2f$-configuration onto the focal plane $P_{2f}$, which projects the Fourier transform of the front focal plane $P_0$ onto $P_{2f}$. We compensate aberrations from non-perfect optical elements, including the spatial light modulator itself, by a Shack-Hartmann wavefront correction algorithm [15]. The sculpted intensity is measured with a CCD camera mounted on a linear translation stage in several target planes $P_j$, covering $\Delta z \in (-12, 12)$mm around the focal plane $P_{2f}$.

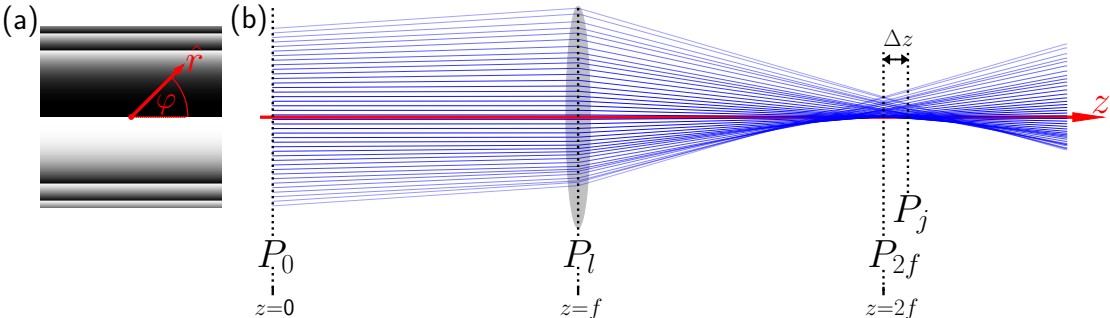

Figure 1: Working principle of the setup with the spatial light modulator located at $P_0$. Cylindrical coordinate system in red ($z$-axis coincides with optical axis). (a) cubic phase pattern displayed on the spatial light modulator to control the beam around the back focal plane $P_{2f}$ sampled at $P_j$ to form an Airy beam. (b) ray simulation of $2f$-setup with phase (a) applied.

The complex transfer functions of Fourier optics provide a full description of linear optical systems [16]. Based on this foundation, phase retrieval algorithms calculate a two-dimensional phase corresponding to a target intensity distribution for a given incident field [17]. To obtain intensity control over a single target plane $P_j$ the phase $\phi_{\mathrm{SLM}}$ is optimized by iterative projection between the incident plane and the target plane. Applying constraints in the target plane $P_{2f}$ and in the front focal plane $P_0$ guides the optimization towards the target intensity. These constraints are implied by the available intensity and the desired target intensity. However, the solutions are not necessarily unique.

Describing the propagation characteristics of an optical beam in a finite volume requires volumetric intensity information. We obtain this information by sampling the

67 beam's intensity at discrete planes $P_j$ around the focal plane $P_{2f}$. The phase $\phi_{\mathrm{SLM}}$ is
68 then calculated with a Gerchberg-Saxton based phase retrieval algorithm [13] (see Figure
69 3). An important subtlety of this algorithm design is that there is no cross-talk between
70 adjacent target planes $P_j$ and $P_{j+1}$. Hence, in each iteration the algorithm solves for all
71 $P_j$ individually and performs a weighted average on the back projected fields at $P_0$. This
72 may lead to a randomly evolving intra-plane intensity [13], which is not suitable for the
73 creation of optical beams.

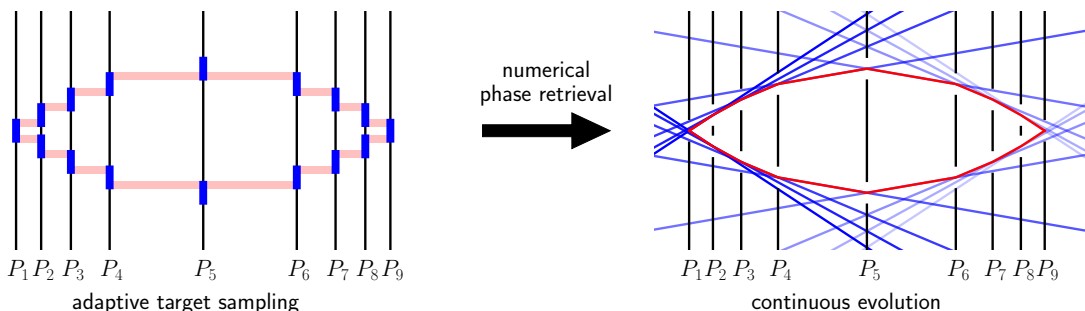

Figure 2: Adaptive real-space target sampling: Creating an overlap between adjacent
target planes to avoid random evolution. Solid black lines indicate the sample planes $P_j$
with binary target (blue). The sample planes are distanced such that adjacent planes
share some overlap (shaded red). The numerical phase retrieval algorithm is guided to the
continuous structure (solid red line).

74 Realizing continuously evolving patterns requires an adjusted target design compen-
75 sating the algorithms mentioned behavior. We have found that the random evolution
76 between adjacent planes can be removed by a proper target sampling. A great discrep-
77 ancy in the target beam profile between adjacent planes result in ambiguous solutions for
78 the intra-plane field. Hence, choosing an adaptive real-space target sampling, tailored to
79 the requested beam, guides the algorithm to converge towards a continuous solution.

80 To influence the optimization as discussed, we choose the target beam sampling such
81 that the intensity at sample plane $P_j$ propagated to $P_{j+1}$ and the intensity at $P_{j+1}$ share
82 an overlap. However, this requirement is not yet strict enough: we have found that we
83 specifically need to create the overlap at the edge of the beam profile. Intuitively, this
84 can be understood as a series of apertures so closely stacked, that the individual rays
85 form the desired contour. Figure 2 illustrates this concept. A two-dimensional bottle
86 beam is formed from a small number of binary beam samples. Ensuring an overlap at
87 the beams edge between adjacent sample planes leads to unambiguous paths for the intra-
88 plane field. This intuitive geometric interpretation also serves to determine the required
89 minimal number of sample planes $N$ and their positions $z_j$. Of course the target beam
90 could be sampled at a much higher rate. Deducing the minimal required $N$ optimizes the
91 computational complexity while still ensuring continuous beam evolution.

92 A common issue with numerical optimization in general is the stagnation in local
93 minima, which applies as well to numerical phase retrieval. A well chosen initial field,
94 i.e., an initial phase guess $\phi_{\mathrm{SLM}}^0$, can serve to improve convergence and avoid stagnation.
95 There are multiple approaches to find an initial phase guess, but due to the huge diversity
96 of the considered targets we choose a random superposition algorithm [13]. This algorithm
97 propagates the three-dimensional target intensities $T_j(\vec{r})$ at $z_j$ back to the incident plane
98 $P_0$ and performs a weighted average on the back-propagated fields.

---

1: **procedure** RANDOM SUPERPOSITION ALGORITHM$(z_j, T_j)$

2:     **return** $\phi_{RS} = \sum_{z_j} \mathcal{P}_{2f}^{-1} \mathcal{P}_z^{-1} \sqrt{T_j} \cdot \exp\left[i\phi_j^{\mathrm{rand}}\right]$

3: **procedure** GLOBAL GERCHBERG-SAXTON$(z_j, T_j)$

4:     $\phi^{(0)} \leftarrow \phi_{RS}$                                     ▷ random superposition phase

5:     $E_{P_0}^{(0)} \leftarrow \sqrt{I_{\mathrm{in}}}$                             ▷ initialization of field in $P_0$

6:     $n \leftarrow 0$

7:     **while** $n < n_{\max}$ **do**

8:         $n \leftarrow n + 1$

9:         $E_{P_0}^{(n)} \leftarrow E_{P_0}^{(n-1)} \exp\left[i\phi^{(n-1)}\right]$

10:        **for** $z \in z_j$ **do**

11:            $E_{P_j}^{(n)} \leftarrow \mathcal{P}_z \mathcal{P}_{2f}\left[E_{P_0}^{(n)}\right]$             ▷ propagation to $P_j$

12:            $E_{P_j}^{(n)} \leftarrow \sqrt{T_j} \cdot \exp\left[i \cdot \arg\left(E_{P_j}^{(n)}\right)\phi_n\right]$         ▷ constraint at $P_j$

13:            $E_{P_{0,j}}^{(n)} \leftarrow \mathcal{P}_{2f}^{-1} \mathcal{P}_z^{-1}\left[E_{P_j}^{(n)}\right]$         ▷ back propagation to $P_0$

14:        $\phi^{(n)} \leftarrow \arg\left[\sum_j E_{P_{0,j}}^{(n)}\right]$

15:     **return** $\phi^{(n)}$

---

Figure 3: Pseudo code of the Gerchberg-Saxton phase retrieval algorithm adopted from [13]. The target intensities $T_j$ applied as a constraint in line 10 are obtained from the adaptive beam sampling.

## 3 Results

### 3.1 Optical bottle and helix beams: Benchmark

A benchmark for arbitrary three-dimensional beam shaping by numerical phase retrieval is the creation of optical beams for which either analytical or closed-form expressions already exist, without actually using this knowledge.

The single-beam optical bottle, for instance, can be realized as a superposition of Laguerre-Gaussian modes [18]. Characteristically, this beam transforms from a bright spot to a homogeneous ring and back to a spot when moving through its focus. Advances in caustic beam engineering have established optical bottles composed of circular auto(de)focusing Airy beams [19] or convex trajectories [9, 20].

As mentioned in the previous section, the number of sampling planes $N$ and their positions need to be derived from the target beam. In order to deduce the number of sample planes $N$ and their positions $z_j$ for an arbitrary intensity map $T(x, y, z)$ consider the intensity overlap $O(\Delta z)$ between two planes axially shifted by $\Delta z$:

$$O\left(\Delta z\right) = \iint_{\mathcal{M}} \mathrm{d}x\mathrm{d}y \ \ T\left(x, y, z_i\right) T\left(x, y, z_i + \Delta z\right) \tag{1}$$

where $T(z)$ is the normalized target beam intensity at the axial position $z$ and $\mathcal{M}$ denotes the focal volume. The steepest descent of $O(\Delta z)$ takes place at

$$\left.\frac{\partial^2 O(\Delta z)}{\partial(\Delta z)^2}\right|_{\Delta z_i} \overset{!}{=} 0 \tag{2}$$

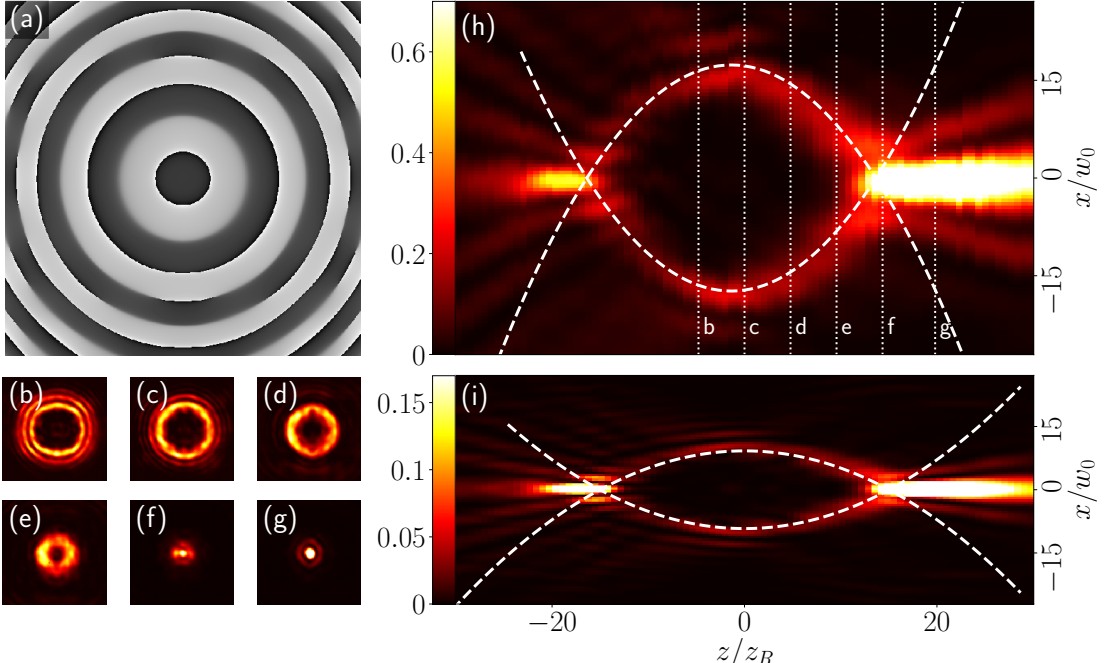

Figure 4: Experimental and numerical results for a single-beam optical bottle. (a) numerically obtained phase pattern, (b)-(g) transverse intensity at the planes indicated in (h) (dotted lines). (h) intensity in the $y$=0-plane including the pre-designed theoretical shape (dashed lines) and its numerically simulated counterpart (i).

115 The vanishing second derivative of $O(\Delta z)$ with respect to the axial shift $\Delta z$ defines the
116 location of the next plane $z_{i+1} = z_i + \Delta z_i$. Applying this procedure iteratively defines all
117 sampling planes $z_j$ and the minimal required number of sample planes $N_{\min}$.

118     The bottle beam's annulus cross-section evolves on a spheroidal trajectory, given in
119 polar coordinates by

$$r(z) = \sqrt{(r_{\max} - r_0)^2 - (z - \bar{z})^2} - r_0. \tag{3}$$

120 Here, $r_{\max}$ denotes the maximal radius of the bottle beam centered at $z = \bar{z}$, while
121 $r_0$ is a radial offset. The length $L$ and the maximal radius $r_{\max}$ are the bottle beams
122 characteristic parameters. Hence, we choose the radial offset $r_0$ such that $r(\pm L/2) = 0$.
123 Consequently, the center of the spheroidal surface is located at $(r_0, \bar{z})$. Describing the
124 bottle beam by a cylindrical symmetric Gaussian of width $w$ moving on the trajectory
125 paramtrized by equation 3 provides the full three-dimensional target beam. Applying the
126 overlap criterion in equation 1 to the three-dimensional target beam yields the number of
127 adaptive sample planes and their positions.

128     Starting from $\phi_{\mathrm{SLM}}^0$ the phase retrieval algorithm calculates the phase pattern $\phi_{\mathrm{SLM}}$ in
129 Figure 4(a). The experimental measured and numerically simulated beam is also depicted
130 in Figure 4. This bottle beam particular bottle beam was recovered from $N = 15$ sampling
131 planes, having a maximal diameter of $2 \cdot r_{\max} = 220\mu m$ and a length of $L = 10\mathrm{mm}$.

132     As desired, the created bottle beam encloses a volume void of any light and the pre-
133 designed trajectory matches the experimental data. The achieved contrast between the
134 bottles surface and its inner region is suitable for manipulation and trapping applications.
135 Apart from a weak intensity asymmetry ($z \leftrightarrow -z$) our result is consistent with bottle
136 beams created from caustic engineering [9]. The creation of various bottle beams within
137 a feasible parameter space ($L \in (14, 54)\, z_R$, $r_{\max} \in (5, 13)\, w_0$, maximal aspect ratio 80:1,
138 where $z_R$ and $w_0$ denote the Rayleigh length and waist of the unmodulated beam) offers
139 a first impression of the flexibility of the presented approach.

140     As a second benchmark, we consider the double-helix point spread function commonly
141 used in super-resolution microscopy [4]. Similar to the optical bottle beam the double helix
142 point spread function can also be described and created by a superposition of Laguerre-
143 Gaussian modes [21] or Bessel beams [22].

144     Since this pattern deviates substantially from the bottle beam discussed earlier, we
145 need to deduce $N$ and the $z_j$ again. The two Gaussian spots are designed to rotate rigidly
146 on a helical trajectory $r(z) = r_{\mathrm{rot}} = $ const, which implies equidistantly spaced $z_j$ along
147 the pattern length $L$ which coincides with the sampling deduced from condition 2.

148     The phase pattern obtained from the numerical phase retrieval is shown in Figure
149 5(b). It shows very similar structures to the analytical phase of the Laguerre-Gaussian
150 superposition [3]. Most intensity of the helix beam is concentrated in the two Gaussian
151 spots. Figures 5(b)-(g) depict the rigid rotation of the equidistant spots. The entire
152 beam propagates shape-invariant throughout the considered volume. Notably our result
153 is obtained without an initial phase guess assuming a Laguerre-Gaussian superposition.

154     The investigated helix beam can be classified as a beam with radially self-accelerating
155 intensity [23]. Hence, there exist a rotating reference frame, in which the beam propagates
156 in a quasi-nondiffractive way. Nondiffractive beams are resilient to small perturbations
157 [23, 24]. This is valid for perturbations smaller or comparably sized to the characteristic
158 beam size, which is the Gaussian spots' waist in our case. Due to their robustness such
159 beams are suitable for many applications where propagation does not take place in vacuum.
160 To prove the quasi-nondiffractive nature of the helix beam, we verify the self-healing after
161 an opaque obstacle. The self-healing properties of the generated beam are tested by a
162 small opaque object placed in the beam path to block one of the two rotating spots near
163 the first target plane $z_0$. The original beam profile was recovered shortly after the obstacle.

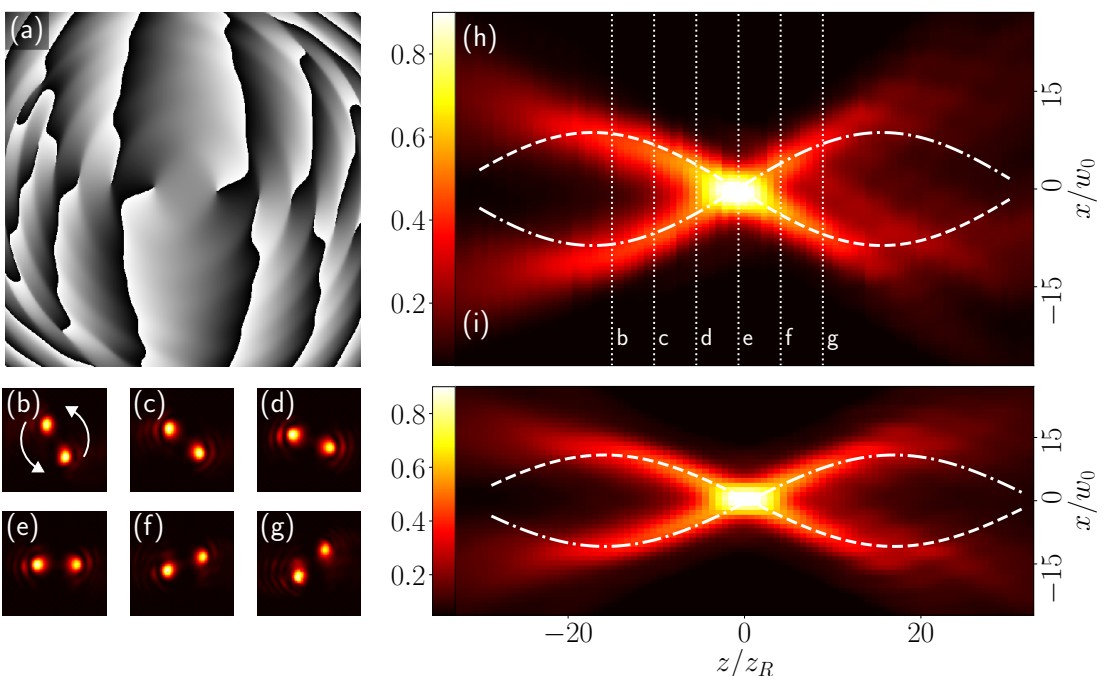

Figure 5: Experimental and numerical results for two Gaussian spots rotating rigidly on a helical trajectory covering a total rotation of $\Delta\varphi = \pi$: (a) calculated phase pattern for counter-clockwise rotation, (b)-(g) transverse intensity at planes indicated with dotted lines in (h). (h) Experimental integrated intensity $\int I(x,y,z)\mathrm{d}y$ and (i) numerical counterpart with theoretical trajectory projected onto the $y = 0$ plane (dashed and dashdotted lines).

The presented results show that our numerical approach combined with an adaptive target sampling is capable of complex beam reconstructions, even when starting from a randomized initial phase guess. The overlap condition implied by equation 2 provides a proper sampling to overcome the random evolution between discrete sampling planes leading to continuously evolving beams. Moreover, it is possible to reproduce beams that exhibit quasi-nondiffractive propagation. Transverse and longitudinal scaling of the created beams can be easily achieved by altering the target beam profile.

## 3.2 Realizing arbitrary beam shaping in three dimensions

We now show that numerical phase retrieval combined with adaptive target sampling provides access to arbitrary three-dimensional intensity sculpting. Not being bound by analytic expressions enables us to create new types of beams with tailored propagation and symmetry properties.

The creation of optical bottles with the discussed analytic approaches commonly exploits its cylindrical symmetry, solving for a trajectory $r(z)$ to calculate a phase $\phi_{\mathrm{SLM}}(r)$ [9]. After the benchmarks in the previous section we go a step further and create a structured intensity surface of the optical bottle beam, that does not obey cylindrical symmetry. To accomplish this we explicitly do not use the bottle beam phase as an initial guess but instead we design a new target beam with the desired properties and apply the phase retrieval algorithm to the adaptively sampled target. The designed surface is structured with a periodic azimuthal intensity gradient and still envelopes a volume of

vanishing intensity. It is possible to create this type of beam with our approach. However, the created azimuthal intensity gradient is of static nature, meaning it does not change when moving through the focus. Additionally adding a rotation to the azimuthal gradient also breaks the symmetry with respect to the focal plane. Although the intensity gradient rotates similarly to the Gaussian spots of the helix beam, these are different types of beams. The spheroidal surface beam emerges from a bright spot, forms a structured annulus and collapses again into a spot, while the rotating helix beam propagates shape invariant throughout all $P_j$. The number and position of sample planes derived from the overlap condition are nearly identical to original bottle beam. The target sampling is only marginally adjusted since the additional rotation is already sampled sufficiently by the bottle beam planes.

A typical result for an optical beam with a rigidly rotating structured spheroidal surface is shown in Figure 6. As demanded, the beam exhibits a periodic azimuthal structure that rotates during propagation. Figure 6(h) illustrates the evolution of the beam profile, which is still continuous despite the substantial complexity increase compared to the benchmark beams. The requested symmetry properties are also fulfilled. Being capable of shaping a beam to this extent separates our approach from techniques that exploit the beam symmetry for simplifying assumptions.

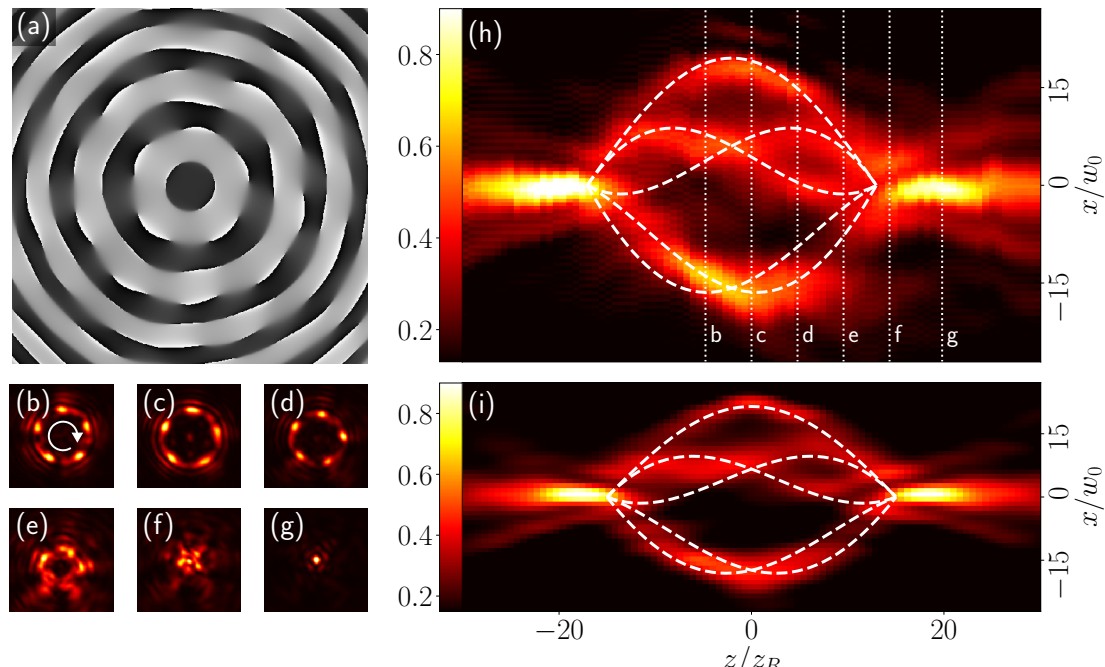

Figure 6: Experimental and numerical results for a single-beam optical bottle with a rotating periodic transverse intensity gradient. (a) numerically obtained phase pattern, (b)-(g) transverse intensity at the planes indicated in (h) (dotted lines) and indicated rotation in (b). (h) Experimental integrated intensity $\int I(x, y, z)\mathrm{d}y$ and (i) numerical counterpart with the theoretical trajectories of the intensity maxima projected onto the $y = 0$ plane (dashed lines).

The second example is a generalization of the double-helix beam. It is known that altering the individual contributions of a Lagerre-Gaussian superposition yields different rotation rates $\frac{\partial\varphi}{\partial z}$ and beam profiles [3, 21]. Yet, the Gaussian spots of the double-helix

point spread function propagate on a trajectory with a circular cross-section (see Figure
5). We now demonstrate that we can vary this cross-section from a circle to a polygon
going beyond the Laguerre-Gauss superposition. As the trajectory (along the $z$–direction)
of the Gaussian spots composing the intensity pattern is no longer rotational symmetric
around the optical axis, the distance between the two Gaussian spots changes with the
propagation distance. Due to its application in super-resolution microscopy the rotation
rate of the double helix point spread function is usually fixed to $\frac{\partial \varphi}{\partial z} = \frac{\pi}{L}$. Similar to
the Laguerre-Gaussian superposition, we can continuously adjust this rotation rate. To
show this we increase the rotation rate by a factor of two, in addition to the varied cross-
section. Regarding the target sampling, we describe the polygonial cross-section in polar
coordinates, which leads to

$$r(\varphi) = r_{\max} \cdot \frac{\cos\left(\frac{\pi}{n}\right)}{\cos\left(\varphi - \frac{2\pi}{n}\lfloor \frac{n\varphi+\pi}{2\pi}\rfloor\right)} \tag{4}$$

where $n$ denoted the polygon order. The target beam is then created from two Gaussian
spots propagating on the trajectory given by $(r\left[\varphi(z)\right]), \varphi(z))^T$.

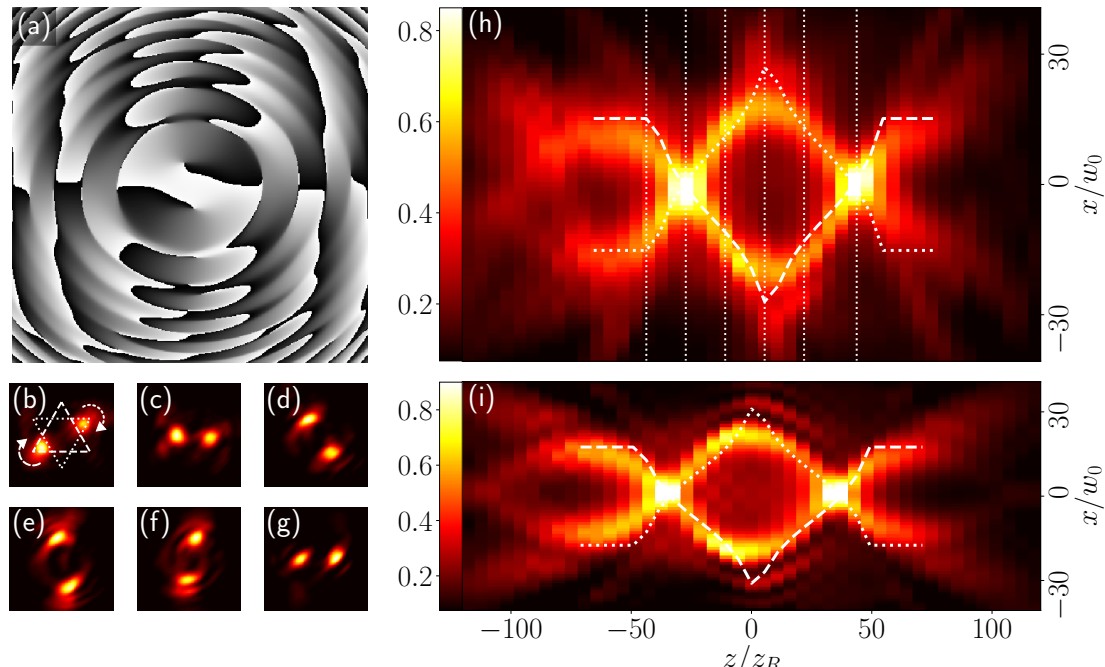

Figure 7: Experimental and numerical results for two Gaussian spots rotating rigidly
on a triangle trajectory covering a total rotation angle of $\Delta\varphi = 2\pi$: (a) calculated
phase for a clockwise rotation, (b)-(g) transverse intensity at planes indicated in (h) with
the pre-designed triangular cross-section in (b). (h) experimental integrated intensity
$\int I(x, y, z)\mathrm{d}y$ and (i) numerical counterpart with theoretical trajectory projected onto the
$y = 0$ plane (dashed and dotted lines).

Typical results for a pair of spots moving on a triangular trajectory are shown in Figure
7. Again the experimental measurements in Figure 7(h) coincides with the numerical
simulations, in Figure 7(i), and the varying distance between the two Gaussian spots can
be observed clearly. Due to the increased rotation rate a full period of the circulation
around the optical axis is visible now. The challenging sections of this beam are located

at the corners of the polygon. The sampling implied by the overlap condition does not deviate significantly from an equidistant sampling. Hence, a equidistant sampling was employed. Still the spots' propagation around the polygons corners suffices to recognize the altered cross-section. As well as the structured intensity surface, this beam serves very well to highlight the performance and functionality of our approach compared to established techniques. The additional effort associated with altering the cross-section and rotation rate is negligible compared to the creation of conventional helix beams.

In order to assess the overall pattern quality of the presented beams more quantitatively, consider the global mean square error $\bar{\varepsilon}$ and the patterns mean diffraction efficiency $\bar{\eta}$ defined by

$$\bar{\eta} = \frac{1}{N} \sum_{\forall z_j} \eta(z_j) N_{\mathcal{S}}(z_j) \quad \text{with} \quad \eta(z_j) = \frac{\sum_{i \in \mathcal{S}} I_i^{\text{act}}(z_j)}{\sum_{i \in \mathcal{M}} I_i^{\text{act}}(z_j)},$$

$$\bar{\varepsilon} = \frac{1}{N} \sum_{\forall z_j} \varepsilon(z_j) N_{\mathcal{S}}(z_j) \quad \text{with} \quad \varepsilon(z_j) = \sum_{i \in \mathcal{S}} \left( \frac{I_i^{\text{act}}(z_j) - I_i^{\text{des}}(z_j)}{I_i^{\text{des}}(z_j)} \right)^2,$$

(5)

where $\mathcal{S}$ and $\mathcal{M}$ represent the signal region and the complete focal region for one axial position $z \in z_j$ with $N_{\mathcal{S}}(z_j)$ being the number of sample points in the signal region of plane $z_j$. $I^{\text{act}}$ and $I^{\text{des}}$ denote the actual and desired intensity. The signal region is defined by the $1/e^2$ contour of the target beam. To prevent the experimental setup from biasing the results and due to the coincidence between the simulated and the measured intensities, we compare the simulated data to the designed target intensities. Applying the metrics of equation 5, a clear differentiation between the helix and bottle beams becomes evident. The diffraction efficiency of the helix beams lies consistently below the bottle beams. Also the mean square error is higher for the helix patterns. Requiring the light to be focused along a point-like three-dimensional trajectory implies much stricter constraints to the light field then being distributed over a specific cross-section like a bottle beam [25], resulting in a reduced $\bar{\eta}$. Remarkably, applying the surface structure to the bottle in Figure 6 does not harm either diffraction efficiency or the mean squared error significantly. For the helix beam on a triangular trajectory however, an increasing $\bar{\varepsilon}$ can be observed, which is closely connected to the sharp corners. The deviation from the designed trajectory in these regions was already visible in Figure 7.

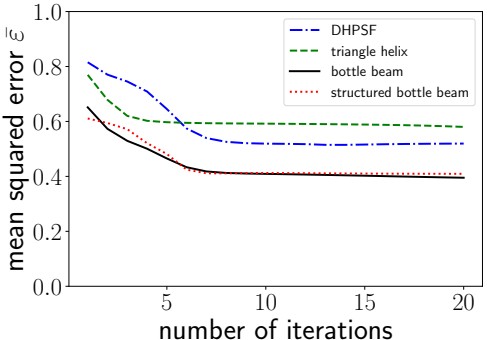 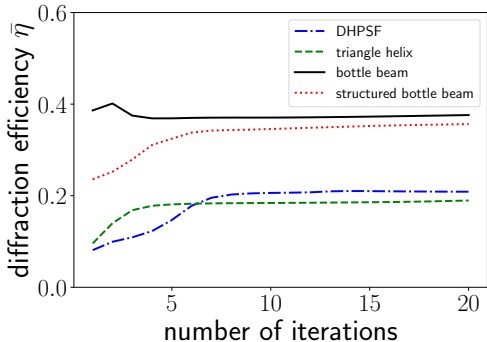

Figure 8: Global mean square error $\bar{\varepsilon}$ and mean diffraction efficiency $\bar{\eta}$ versus the number of iterations. The number of sampling planes is fixed at $N = N_{\text{min}}$ for all patterns presented in this paper (DHPSF in Figure 5, triangle helix in Figure 7, bottle beam in Figure 4 and structured bottle beam in Figure 6).

The presented beam shaping operations should be understood as examples, representing only a subset of potential diversification. All patterns created in this paper show that a proper target sampling is key to obtain continuously evolving optical beams when using numerical phase retrieval in three dimensions sampled at multiple axially shifted planes. Although the considered three-dimensional beam profiles are of complex nature, numerical simulation and experimental measurements coincide remarkably well, emphasizing the achievable predictability and control over the beam propagation.

# 4 Quantitative evaluation of equidistant target sampling

Considering each target beam profile individually and applying the introduced overlap criterion helped to deduce the two critical parameters of adaptive target sampling: the number of sample planes $N$ and their positions $z_j$. In the previous section we demonstrated the beam shaping capabilities that can be achieved using such an adaptive target sampling. Here we investigate the influence of these parameters separately. Looking at the results in Figures 4, 5, 6 and 7, the experimental measurement and theoretical simulation coincide well. However, experimental data can be flawed by several effects like finite diffraction efficiency, phase mask aliasing. To account for this we conduct our investigations based on the theoretical simulations of optical bottle beams. Please note, that these will also include the finite active region, and resolution of the used spatial light modulator. Following the previous calculations, we find $N_{\min} = 16$ for the considered bottle beam. The simulated results for this specific bottle beam sampled with different $N \in \{N_1 = \frac{1}{2}N_{\min}, N_2 = N_{\min}, N_3 = \frac{3}{2}N_{\min}\}$ using either adaptive or equidistant sampling are depicted in Figure 9.

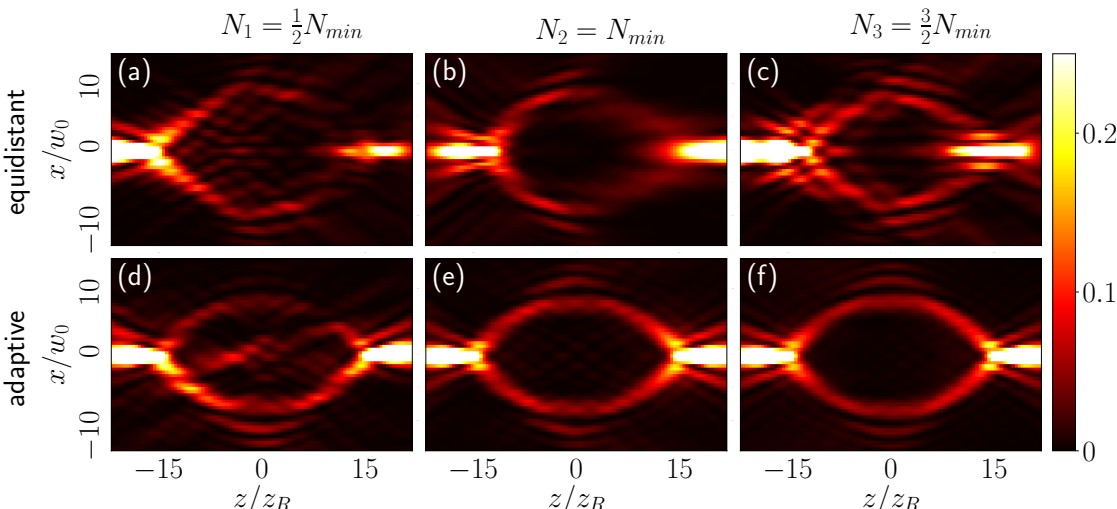

Figure 9: Simulation of the same bottle beam with different sample rates $N_i$. Upper row is obtained from equidistant sampling, while lower row is based on the presented adaptive target sampling.

Despite that all bottle beams are created from the same target, the beneficial influence of adaptive sampling combined with an appropriately sized $N$ is clearly visible. First, consider the equidistant sampled bottle beam in Figure 9(a-c). The bottle beams surface

is composed of mutliple segments of high intensity and exhibits a strong asymmetry for $z \leftrightarrow -z$. For $N = N_{\min}$ these high intensity segments seem to merge into a more continuous surface, but even at the highest sample rate $N_3$ there is no transition to a closed intensity surface like in Figure 9(f). The observed segmentation is most prominent at regions of low/vanishing overlap being the opening and closing points of the bottle beam (see Figure 9(c)). Such an beam propagation would be expected for non proper sampling resulting in ambiguous intra-plane intensities. Arguably these beams created with the equidistant sampling could be considered to be bottle beams, having a higher intensity at the edge then in the center. Due to the degree of discontinuity and the high number of artifacts penetrating the inner volume the applicability of these bottle beams is very restricted. For all sampling rates depicted in Figure 9 the pattern quality of the adaptive sampled bottle beams exceeds the equidistant sampled bottle beams. Also its surface homogeneity improves significantly when increasing number of sample planes $N \geq N_{\min}$. Notably the trajectory $r(z)$ in equation 3 can already be identified even at the lowest sampling rate $N_1$. In Figure 9(d-f) the transition from an discontinuous surface with artifacts in the inner volume to a true bottle beam enclosing a volume is observed around $N \approx N_{\min}$.

Investigating the diffraction efficiency and mean square error of an optical bottle beam for different number of adaptive sample planes yields the results in Figure 10. To add more consistency to these results we used the same initial phase guess since it is generated with some random phase offset between the target planes. The transition observed in 9(d, e, f) is also present in Figure 10. Around $N \approx N_{\min}$ the improvement gained from a higher number of sample planes stagnates to $\frac{\partial \bar{\eta}}{\partial N} = \frac{9.56 \cdot 10^{-4}}{\text{sample plane}}$ and $\frac{\partial \bar{\varepsilon}}{\partial N} = -\frac{5.71 \cdot 10^{-4}}{\text{sample plane}}$. Hence, increasing the number of sample planes above the minimal required $N \geq N_{\min}$ effects the diffraction efficiency and the mean square error only marginally compared to the additional computation time. However, the idea of a multi-layer design algorithm based on two-dimensional Fourier transforms would become obsolete by increasing the number of sample planes to comparable values of other approaches build around three-dimensional Fourier transformations and sampling on a three-dimensional grid with $\mathcal{O}(N) = 100$.

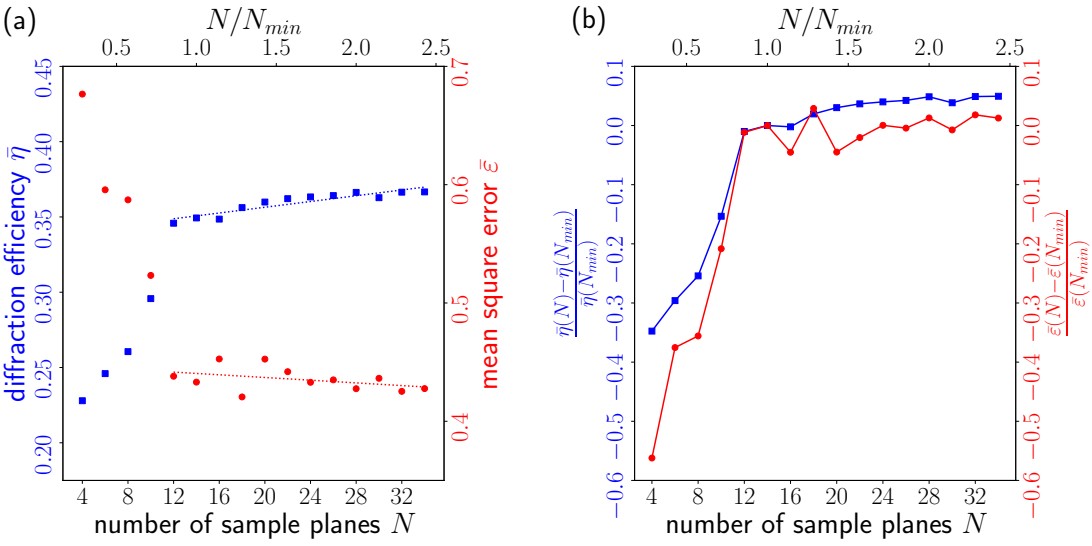

Figure 10: Mean diffraction efficiency $\bar{\eta}$ and global mean squared error $\bar{\varepsilon}$ versus the number of sampling planes when employing adaptive target sampling. (a) absolute values with a linear fit for $N > N_{\min}$ and (b) relative improvements compared to $N = N_{\min}$.

Summarizing the evaluation of the adaptive target sampling applied to the bottle beam by means of the error metrics in equation 5 and their beam profile in Figures 9 and 10, the proposed method yields the expected improvements. We have shown that adaptive sampling enhances the overall pattern quality. Additionally the minimal number of sampling planes was calculated based on the target pattern and reproduced by numerical simulation. As intended the value of $N_{\min}$ marks a characteristic point for the algorithms convergence. For $N$ exceeding $N_{\min}$ only minor improvements take place. Hence, $N_{\min}$ can be considered the optimum number of sample planes for a minimal algorithm running time.

# 5  Conclusion and Outlook

In this paper, we have shown that the three-dimensional intensity distributions of complex beams can be created by means of multi layer numerical phase retrieval. Despite the multi-layer sampling the obtained beam profiles evolve continuously throughout the focal volume. Our approach is capable of producing these optical beams with pre-designed non-trivially evolving transverse profiles without sacrificing the patterns fidelity. We have shown that our approach can reproduce light patterns of different approaches.In addition, we have successfully created new complex beams that have not been generated by conventional techniques, showcasing the considerable sculpting possibilities of our approach.

The requested target beam properties can be directly applied in real-space targets instead of tracking down their origin to the original beam or the generating phase pattern. These large degrees of freedom increase the applicability of advanced tailored optical fields. Furthermore, dynamic manipulation can be achieved by sequences of phase patterns only limited by the spatial light modulators pixel refresh rate. The remaining intensity inhomogeneities along the propagation trajectory may be compensated by additional amplitude control of the incident field [20].

Numerical phase retrieval for three-dimensional beam shaping may open the door to novel optical potentials built on top of already existing classes of optical beams. In the future our method could help to launch new developments in various fields: quantum gases confined to spatially curved potentials, particle manipulation and guiding along arbitrary trajectories or laser writing of new types of structures could be achieved adopting our approach. Given the flexibility and simplicity of the presented approach, it may be a valuable tool for applications, wherever precisely controlled optical potentials are essential.

# Acknowledgements

**Funding information**   This work has been supported by the Alexander-von-Humboldt Stiftung, DFG (SFB/TR 185 project A2), and funded by the Deutsche Forschungsgemeinschaft (DFG, German Research Foundation) under Germany's Excellence Strategy – Cluster of Excellence Matter and Light for Quantum Computing (ML4Q) EXC 2004/1 – 390534769.

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
