# Peer review of "Three-dimensional sculpting of laser beams"

_SciPost Physics_

## Round 1 · Referee Report · Anonymous (Referee 2) · 2020-5-12

Strengths

1- Seems like a very useful method for any application where the output of interest is the light intensity and where the phase is unimportant, this means it should find application in optical tweezers, ultra-cold atoms, and laser engraving. 2- Show the versatility of the result by realizing four optical beams two of which have not been previously realized and which are not a simple superposition of fundamental modes.

Weaknesses

1- I was expecting a general formula for the most optimal selection of the plane location for a general intensity target map I(x,y,z), I cannot help but think there must be a way to formalize this process by looking at the maximum of the dI(x,y,z)/dz gradient in the plane where z is the beam propagation direction. And spacing the planes such that the expected maximum expected to change intensity between two planes is constant. 2- The complex optical demonstrations are nice, but I personally thought a comparison with other existing methods to create these 3D light profiles would have been nice. Especially a comparison of the error between the target intensity for the different methods and the achieved intensity. The reason this seems relevant to me is that I have no clue how much of a problem "randomly evolving intra-plane intensity [11]" are and how well this method performs compared to previous methods especially considering that complexity is added to the problem by having to figure out analytically for each different beam figure out the optimal spacing between the planes.

Report

In general, I found the quality of results in "Three-dimensional sculpting of laser beams" to be good. This paper demonstrates the use of optimized plane selection in the creation of 3D optical beams using a 2D phase shaping SLM. It qualitatively describes the method used to select the planes and clarifies through the use of four realized optical beams two of which have not been previously realized which are not a simple superposition of fundamental modes. Their results will be applicable to any application where 3D beam shaping of the light intensity is desirable such as cold atoms. I would recommend the publication of this paper in SciPost Physics after the following revisions and comments have been addressed.

Requested changes

1- For Figure 3-6 (h),(i), it would be easier to compare the experimental and the theoretical images if the vertical extent of the theoretical and experimental were the same. In addition, for the (i) panel I would avoid using the jet colour scale since it tends to be the colour scheme most likely to not be visible for colour blinds. Finally, the purpose of plots (i) and (h) is to be compared so I would suggest using the same colour scheme with the same range on the colour bar due to no extra information being communicated from using two different colour schemes with different scales, in fact, I would argue that information is lost in this case. 2- You mention at the beginning of the paper that your method performs better than the Gerchberg-Saxton based phase retrieval algorithm used on equally space planes especially for the intra-plane intensity, but nowhere in the paper is your method quantitatively compared. I think it would be interesting if you used the Gerchberg-Saxton based phase retrieval algorithm with equally space planes to create the same patterns you created using your method and compared the results numerically by calculating the error between the experimental pattern and the theoretical generated intensity pattern. This would illuminate how much you are gaining from the extra complexity of having to determine the right sampling plane. 3- I was expecting a general formula for the most optimal selection of the plane location for a general intensity target map I(x,y,z), I cannot help but think there must be a way to formalize this process by looking at the minimum of the dI(x,y,z)/dz gradient in the plane where z is the beam propagation direction. Then spacing the planes by \Delta z = I(x,y,z)/|dI(x,y,z)/dz| where this is calculated at the location of steepest descent. The reason I think this is important is that the novel idea of this paper is the selection of none uniformly space plane and I think it would add a lot of value to the paper to have a more rigorous and mathematical criterion for the selection of these planes. 4- On line 93, "[…] are multiple approaches to a find an initial phase guess […]" should read "[…] are multiple approaches to find an initial phase guess […]". 5- On line 174, "To accomplish this we do explicitly not use the bottle beam phase as an initial guess but […]" should read "To accomplish this we do not explicitly use the bottle beam phase as an initial guess but […]" 6- Figure 4(h),(i) caption mentions dashed and dashed dot line but only dashed lines are present. 7- On line 214, "[…] simulations 6(i) and the varying […]" should read "[…] simulations, in Fig. 6(i), and the varying […]"

  • validity: high
  • significance: good
  • originality: good
  • clarity: good
  • formatting: reasonable
  • grammar: good

Author:  Michael Köhl  on 2022-09-09  [id 2804]

(in reply to Report 2 on 2020-05-12)

i. For Figure 3-6 (h),(i), it would be easier to compare the experimental and the theoretical images if the vertical extent of the theoretical and experimental were the same. In addition, for the (i) panel I would avoid using the jet colour scale since it tends to be the colour scheme most likely to not be visible for colour blinds. Finally, the purpose of plots (i) and (h) is to be compared so I would suggest using the same colour scheme with the same range on the colour bar due to no extra information being communicated from using two different colour schemes with different scales, in fact, I would argue that information is lost in this case.
A: We have adjusted the respective figures to show the same colour scheme and range, except for the optical bottle beam. Since the bottle beam exhibits a strong focusing at the opening/closing points we reduced the color range s.t. the high intensity surface (especially for small |z|) is still visible.

ii. You mention at the beginning of the paper that your method performs better than the Gerchberg-Saxton based phase retrieval algorithm used on equally space planes especially for the intra-plane intensity, but nowhere in the paper is your method quantitatively compared. I think it would be interesting if you used the Gerchberg-Saxton based phase retrieval algorithm with equally space planes to create the same patterns you created using your method and compared the results numerically by calculating the error between the experimental pattern and the theoretical generated intensity pattern. This would illuminate how much you are gaining from the extra complexity of having to determine the right sampling plane.
A: We have included a section where we compare the results of the equidistant with the adaptive sampling. Since the two key properties of our methods consist of the minimal number of sample planes and their positions we compared the results for three different number of planes and for the equidistant/adaptive sampling. One can observe the beneficial influence of the adaptive target sampling combined with a proper number of sampling planes. We think that the results in the added figure are unambiguous concerning the pattern quality improvement when using the adaptive target sampling.
Regarding the error evaluation: We included a plot displaying the mean square error (MSE) and the diffraction efficiency (DE) vs. number of iterations for all created types of beams. The discrepancy between the helix and bottle beams is consistent with the literature (see ref[25]). Another plot to illuminate the influence of the number of sample planes on DE and MSE was included.
A few comments on this type of error evaluation in the context of the used algorithm:
There are three intensity distributions that could be considered: the target T, the simulated intensity S and the intensity measured in the experiment E. As the already included figures indicate: E and S show good coincidence. The remaining deviation would probably be dominated by the deficits of the used setup which are not included in the simulation (finite NA, the SLMs finite diffraction efficiency, aliasing and pixel cross talk). The other possible comparison (which was also conducted in the new manuscript) would be to consider the error between T and S. The problem here is that there are no physics involved in the design of T. Considering the limitations of the setup (e.g. the NA) helps to narrow down the possible beam shapes, but the designed pattern is not necessarily realizable in the first place. The MSE could therefore also be impacted by the target beam model and not only by the algorithms performance/convergence.

iii. I was expecting a general formula for the most optimal selection of the plane location for a general intensity target map I(x,y,z), I cannot help but think there must be a way to formalize this process by looking at the minimum of the dI(x,y,z)/dz gradient in the plane where z is the beam propagation direction. Then spacing the planes by \Delta z = I(x,y,z)/|dI(x,y,z)/dz| where this is calculated at the location of steepest descent. The reason I think this is important is that the novel idea of this paper is the selection of none uniformly space plane and I think it would add a lot of value to the paper to have a more rigorous and mathematical criterion for the selection of these planes.
A: We found a way to reproduce our specific solutions for the adaptive target sampling by considering the intensity overlap between two axially shifted planes. We used the point of steepest descent of the overlap. This criterion can be applied to any three-dimensional intensity map.
Considering the gradient in propagation direction seems problematic for some three-dimensional beam, where the gradient vanishes at some z on the trajectory. This causes the expression I(x,y,z)/|dI(x,y,z)/dz| to diverge.

---

## Round 1 · Referee Report · Anonymous (Referee 1) · 2020-5-12

Strengths

  1. Demonstration of interesting 3D optical laser beams using a phase-only spatial light modulator.
  2. Clear presentation of results.

Weaknesses

See comments below.

Report

This paper experimentally (i.e. with actual laser beams) demonstrates the generation of a number of 3D light intensity patterns using a a single phase-only spatial light modulator. It marks a significant contribution to this topical area which could prove useful for a number of different applications.

It is clearly worthy of publication but I have a number of questions /suggested changes (listed below) that should be considered first.

Requested changes

(i) Is it true that the only advance in the applied algorithm (for example over ref [11]) is the careful sampling of the target volume? If there are any other changes to the phase retrieval algorithm then these should be made clear.

(ii) The term arbitrary is perhaps too lightly used in a few places - it is not possible to create truly arbitrary 3D optical potentials from a single propagating beam. This should be acknowledged and the word should probably be removed (or qualified) in lines 6, 165 and 167.

(iii) It would be better to have the same axis ranges and colour scales in the experimental /theoretical integrated density plots in figures 3, 4, 5 & 6.

And some smaller points:

(iv) In line 34 it seems a bit strange to say that the intensity between the discrete planes 'evolves randomly' - I think the point is more that it in evolves in an unspecified /uncontrolled way.

(v) Sentence on lines 34-36 probably needs changing regarding the 'truly arbitrary' statement [see (ii) above], but also it would read better if 'progress in order' were replaced with 'step'.

(vi) Line 38 - replace 'an improved numerical phase retrieval' with either 'improved numerical phase retrieval' or 'an improved numerical phase retrieval method'

(vii) Replace 'at' with 'using'.

(viii) Replace 'composes' with 'is composed of'

(ix) Line 79: is the clause 'propagated to $P_{j+1}$' important?

(x) I think there is a 'while' missing before 'still ensuring'

(xi) Line 120 & 121 are somewhat of a repeat of line 94.

(xii) Line 151 - I think should say '... in a quasi-nondiffraction way...'

(xiii) Line 174 should be changed from '...we do explicitly not use...' to '...we explicitly do not use...'

(xiv) Line 183 - should be '... a structured annulus ...'

(xv) Line 241 - 'pattern' should be 'patterns'

(xvi) Line 246 - 'build' should be 'built'

(xvii) Maybe consider adding references to:
Nature volume 561, pages79–82(2018)
PRL 124, 023201 (2020)
(this was just two papers I know related to this topic which weren't referenced, but I am not up to date with the literature).

  • validity: high
  • significance: ok
  • originality: ok
  • clarity: good
  • formatting: excellent
  • grammar: good

Author:  Michael Köhl  on 2022-09-09  [id 2803]

(in reply to Report 1 on 2020-05-12)
Category:
answer to question

i. Is it true that the only advance in the applied algorithm (for example over ref [11]) is the careful sampling of the target volume? If there are any other changes to the phase retrieval algorithm then these should be made clear.
A: As stated in the manuscript we adopted the “Global Gerchberg Saxton algorithm” in ref [13] (previous version red[11]) without adding additional improvements. To achieve the presented beam shaping we have changed the target sampling, i.e. we choose not equidistant target planes.

ii. The term arbitrary is perhaps too lightly used in a few places - it is not possible to create truly arbitrary 3D optical potentials from a single propagating beam. This should be acknowledged and the word should probably be removed (or qualified) in lines 6, 165 and 167.
A: We agree with the referee that that one cannot create truly arbitrary beams entirely from a single beam. The phrase “arbitrary” is used frequently in the referenced Literature [e.g 7,8,9] to put emphasis on the large variety of beams created (different cross-sections and trajectories). Since we created multiple beams with inherently different properties, this phrase is used in the same sense as in the referenced Literature.

iii. It would be better to have the same axis ranges and colour scales in the experimental /theoretical integrated density plots in figures 3, 4, 5 & 6.
A: We have adjusted the respective figures to show the same colour scheme and range, except for the optical bottle beam. Since the bottle beam exhibits a strong focusing at the opening/closing points we reduced the color range s.t. the high intensity surface (especially for small |z|) is still visible.

---

## Round 1 · Referee Report · Anonymous (Referee 3) · 2020-5-13

Strengths

1. The generation of accurate three-dimensional light intensity distributions finds application in many fields, including super resolution microscopy, holographic optical tweezers, cold atom experiments, or information encoding in optical communication.
2. The capabilities of their approach are easily captured by their choice of examples.
3. The agreement between the measured intensity distributions and the numerical calculations is remarkable.

Weaknesses

1. Literature in the field is not properly referenced and a comparison of their approach to other well-known methods is never made.
2. Some statements are misleading or inaccurate.

Report

In their manuscript, Kree and Köhl, report on the generation of continuous three-dimensional light intensity distributions using a numerical phase retrieval algorithm. By using a three-dimensional extension of the celebrated Gerchberg-Saxton (GS) algorithm and a proper multiplane sampling, they obtain optical bottle beams of tuneable sizes and different types of helix beams. The measured intensity distributions are presented along with the results predicted by the numerical algorithm.

Shaping light into arbitrary 3D configurations allowed by diffraction is a key technology in many fields, including microscopy, lithography, or optical communications. Combined with spatial light modulators, the ability to obtain continuous light intensity distributions can be used, e.g., for laser trapping of cold atoms. In this respect, the topic of this paper is interesting and timely.

Reading the manuscript one could get the impression that continuous three-dimensional shaping of light beams had not been accomplished before. However, there are a number of works that demonstrated 3D shaping using numerical phase retrieval algorithms. Some examples are:

1. Experimental demonstration of holographic three dimensional light shaping using a Gerchberg–Saxton algorithm, G. Whyte and J. Courtial, New J. Phys. 7, 117 (2005). There, a 3D Gerchberg-Saxton algorithm is also used to obtain phase holograms that create complex intensity distributions experimentally.

2. Holographic optical tweezers obtained by using the three-dimensional Gerchberg–Saxton algorithm, Chen et al., J. Opt. 15, 035401 (2013). In this work, intensity tailoring in continuous 3D volumes rather than in multiple discrete two-dimensional planes is used for particle trapping.

The central issue Kree and Köhl aim at solving via adaptive sampling is that in the variant of the GS algorithm they employ to create stacks of 2D light patterns “there is no cross-talk between adjacent target planes” and “the intensity between these discrete planes evolves randomly”. However, previous works have shown that, since the field in one plane completely determines the field everywhere else, it is more convenient to propagate the fields from the Fourier plane to the other axially shifted planes before the constraints are propagated backwards using the inverse FFT. This propagation in free space can be done using the Fresnel transform or the angular spectrum method, as proposed or used, e.g., in:

3. Computer-generated holograms from 3D-objects written on twisted-nematic liquid crystal displays, Haist et al, Opt. Commun. 140, 299 (1997).
4. Interactive application in holographic optical tweezers of a multi-plane Gerchberg-Saxton algorithm for three-dimensional light shaping, Sinclair et al., Opt. Express 12, 1665 (2004).
5. Iterative algorithms for holographic shaping of non-diffracting and self-imaging light beams, Sinclair et al., Opt. Express 12, 5475 (2004).
6. Speckle-suppressed phase-only holographic three-dimensional display based on double-constraint Gerchberg-Saxton algorithm, Chang et al., Appl. Opt. 54, 6994 (2015).
7. Complex amplitudes reconstructed in multiple output planes with a phase-only hologram, Wu et al., J. Opt. 17, 125603 (2015).

In this latter paper, for instance, it is shown that both the amplitude and the phase of the beam can be controlled simultaneously in up to seven planes (to be compared with the six planes used in the examples here) with a phase-only hologram.

None of these works are mentioned in the manuscript. As a consequence, it is difficult to judge how their approach (which consists in just an adaptive sampling of the target intensity in several planes combined with an extension of the GS algorithm) performs and compares to well-established techniques, the potential advantages and disadvantages. In addition, I find that some relevant information about the actual scheme is missing. This includes the running time of the algorithm, the average diffraction efficiency of the solution, the scaling with the number of planes, etc. For these reasons, I cannot support publication in its current form.

Questions/comments:
1. For the iterative GS algorithm they follow Ref. [11]. Since there are already different variants of the GS algorithm for 3D geometries, I would appreciate a reminder of this algorithm in the text. This would make the paper more self-contained.
2. In the text of Section 2 and Fig. 2 they explain that “sampled planes are distanced such that adjacent planes share some overlap”. What does it mean some overlap? This criteria is made clearer in the examples, but should be defined more formally in this section. How is it done for arbitrary shapes if the analytic or closed form of the field is not known? How important is actually this adaptive sampling (as used in the first example) when compared to a fixed spacing sampling with the same number of planes (as in the other examples)?
3. In their setup, they compensate aberrations from non-perfect optical elements, including the spatial light modulator. Why is this needed? Is it critical? How sensitive with respect to aberrations is their method if light is strongly focused with high NA lenses?
4. In Fig. 3 they refer to a weak intensity asymmetry (z -> -z). What is the origin of that?
5. How different are the retrieved holograms compared to other phase patterns calculated by other methods?
6. What is the average diffraction efficiency and how does it scale with the number of sampling planes? What about the running time of the algorithm?

Requested changes

  1. Authors should include relevant works in the reference list and clearly state the contribution of their research in this context.

  2. See other suggestions in the main report.

  • validity: ok
  • significance: low
  • originality: low
  • clarity: ok
  • formatting: good
  • grammar: excellent

Author:  Michael Köhl  on 2022-09-13  [id 2811]

(in reply to Report 3 on 2020-05-13)
Category:
answer to question

We have added three more references (new references 11, 12, and 25).

Answers to the specific questions:

i. The central issue Kree and Köhl aim at solving via adaptive sampling is that in the variant of the GS algorithm they employ to create stacks of 2D light patterns “there is no cross-talk between adjacent target planes” and “the intensity between these discrete planes evolves randomly”. However, previous works have shown that, since the field in one plane completely determines the field everywhere else, it is more convenient to propagate the fields from the Fourier plane to the other axially shifted planes before the constraints are propagated backwards using the inverse FFT.
A: As ref[13] indicated, propagating the field sequentially from one plane to the next may not always lead to a better overall three-dimensional beam (see comparison between Global GS and Sequential GS in ref[13]). Based on the results of ref[13] we choose the Global GS.

ii. Complex amplitudes reconstructed in multiple output planes with a phase-only hologram, Wu et al., J. Opt. 17, 125603 (2015). In this latter paper, for instance, it is shown that both the amplitude and the phase of the beam can be controlled simultaneously in up to seven planes (to be compared with the six planes used in the examples here) with a phase-only hologram.
A: The two features of adaptive target sampling are the number of sample planes N and their positions distributed over the pattern z_j. Both of these are calculated for each pattern individually. Typical numbers for the beams presented in the manuscript are on the order of (15-30). Please note that the depicted cross-sections are not the planes where we employ the intensity constraints. The point of performing the adaptive target sampling is to obtain a continuously evolving beam. Hence, each axial plane serves as a probe for the beam quality. The depicted planes distributed along the controlled volume simply serve to illustrate that the cross-section evolves as demanded.

iii. For the iterative GS algorithm they follow Ref. [11]. Since there are already different variants of the GS algorithm for 3D geometries, I would appreciate a reminder of this algorithm in the text. This would make the paper more self-contained.
A: The pseudo code explaining the algorithm procedure was added to the manuscript.

iv. In the text of Section 2 and Fig. 2 they explain that “sampled planes are distanced such that adjacent planes share some overlap”. What does it mean some overlap? This criteria is made clearer in the examples, but should be defined more formally in this section. How is it done for arbitrary shapes if the analytic or closed form of the field is not known? How important is actually this adaptive sampling (as used in the first example) when compared to a fixed spacing sampling with the same number of planes (as in the other examples)?
A: Please see report02 (iii).

v. In their setup, they compensate aberrations from non-perfect optical elements, including the spatial light modulator. Why is this needed? Is it critical? How sensitive with respect to aberrations is their method if light is strongly focused with high NA lenses?
A: We observed a distortion of the PSF when a constant phase mask was displayed on the SLM. A distorted PSF would also spoil the three-dimensional patterns. We found that the main contribution to the phase aberrations originated in the surface flatness of the SLM. The SH-procedure mainly counters this aberration and defines the focal plane to the position of the camera chip. Assuming that the aberration compensation improves the PSF in the focal volume, we also employed it on the three-dimensional patterns. Additionally the Shack-Hartman procedure allows us to probe the beam that is incident on the SLM (obtain relative positioning, waists, etc.) which is on of the constraints applied in the phase retrieval algorithm at the SLM plane. Hence, the SH-procedure improves the pattern quality. The necessity of aberration correction is obviously directly linked to the quality of the used setup and may not be present for high-end optics.

vi. In Fig. 3 they refer to a weak intensity asymmetry (z -> -z). What is the origin of that?
A: This feature is part of the optimized solution, meaning that the algorithm converged towards this asymmetry. Hence, the origin lies within the applied phase map. This is not a systematic effect. This bottle beam was created from the minimal number of sample planes calculated as described.

vii. How different are the retrieved holograms compared to other phase patterns calculated by other methods?
A: The benchmark beams serve well to investigate this question. Looking at the analytical counterparts of the calculated phase patterns shows that the overall structure is very similar (e.g compare to the DHPSF phase map in ref [3] or the bottle beam in ref[9]).
The desired target in our case is not necessarily a physical field, the opposite is true for the analytic phase maps that are either created from closed form expressions or analytical modes, meaning that there will be some deviations, but they do not affect the beam propagation significantly. Also, due to the numerical optimization and the randomized initial guess there will always be some deviation to the “optimal” solution. Therefore we have not conducted a quantitative evaluation of that question.

viii. What is the average diffraction efficiency and how does it scale with the number of sampling planes? What about the running time of the algorithm?
A: We included an error metric in the manuscript including the MSE and the diffraction efficiency vs. iterations. We found that the type of pattern (surface vs. point-like cross-section) influences these measures the most.
Concerning the running time: The adaptive sampling needs to be deduced before hand. Hence, these calculations are not part of the iterative projection algorithm itself. Assuming that the target sampling is known a-priori the algorithms running time is not affected by any means. Running on an CPU (Intel i5-7200 @ 2.5GHz) takes approx 2s/iteration/plane when the field is zero-padded up to 2048x2048. Of course this time can be decreased significantly by calculating on a proper GPU.

---

## Round 2 · Referee Report · Anonymous · 2022-10-13

Report
I am now satisfied that this manuscript is suitable for publication in SciPost.

---

## Round 2 · Referee Report · Anonymous · 2022-11-22

Report
I am happy that the revised version of the manuscript has address all of my comments and reccomend publication in SciPost.

---

## Round 2 · List of Changes

Changes are marked in red in the manuscript.

You are currently on this page

---

## Editorial Decision

editorial_decision: